# The BMP2/4 ortholog Dpp can function as an inter-organ signal that regulates developmental timing

Linda Setiawan[1], Xueyang Pan[2], Alexis L Woods[1], Michael B O'Connor[2], Iswar K Hariharan[1]

**Developmental transitions are often triggered by a neuroendocrine axis and can be contingent upon multiple organs achieving sufficient growth and maturation. How the neurodendocrine axis senses the size and maturity of peripheral organs is not known. In *Drosophila* larvae, metamorphosis is triggered by a sharp increase in the level of the steroid hormone ecdysone, secreted by the prothoracic gland (PG). Here, we show that the BMP2/4 ortholog Dpp can function as a systemic signal to regulate developmental timing. Dpp from peripheral tissues, mostly imaginal discs, can reach the PG and inhibit ecdysone biosynthesis. As the discs grow, reduced Dpp signaling in the PG is observed, consistent with the possibility that Dpp functions in a checkpoint mechanism that prevents metamorphosis when growth is insufficient. Indeed, upon starvation early in the third larval instar, reducing Dpp signaling in the PG abrogates the critical-weight checkpoint which normally prevents pupariation under these conditions. We suggest that increased local trapping of morphogen within tissues as they grow would reduce circulating levels and hence provide a systemic readout of their growth status.**

## Introduction

Organismal development is often orchestrated by a neuroendocrine axis, such as the hypothalamic–pituitary axis in mammals. Mechanisms likely exist by which the growth and maturation of peripheral organs are monitored by the neuroendocrine axis before important developmental transitions. In most cases, these mechanisms remain undefined. The onset of metamorphosis in *Drosophila* is a dramatic developmental transition which lends itself to genetic analysis (reviewed by Yamanaka et al [2013], Boulan et al [2015]). The larva is capable of feeding and hence acquiring additional nutrients for growth until it achieves its final size. In contrast, the pupa is essentially a closed system where any new growth and tissue remodeling can only occur either by mobilizing stored nutrients or by the breakdown of larval tissues. The mechanisms by which

*Drosophila* larvae assess their growth and maturity before committing to metamorphosis are of considerable interest because they might suggest principles that govern the relationship between growth and developmental timing in diverse organisms.

The endocrine gland that regulates the timing of metamorphosis in *Drosophila* is the ring gland (Fig 1A), which is composed of three main parts (King et al, 1966). The prothoracic gland (PG) secretes the steroid hormone ecdysone, the corpus allatum (CA) secretes juvenile hormone (JH), and the corpora cardiaca (CC) are neurosecretory cells that secrete adipokinetic hormone (AKH) (discussed in Christesen et al [2017]). During the third larval instar, entry into metamorphosis is characterized by a decline in JH levels and a steep increase in the level of ecdysone. In *Drosophila* at least, changes in JH levels do not seem to have a major effect on the timing of pupariation, whereas ecdysone levels seem crucial (Riddiford et al, 2010; Boulan et al, 2015). In the PG, multiple cytochrome P450 enzymes, encoded by the Halloween genes, convert cholesterol to ecdysone (Petryk et al, 2003; Gilbert, 2004; Warren et al, 2004; Ono et al, 2006). Larval molts and the onset of pupariation are each preceded by distinct peaks of circulating ecdysone secreted by the PG. The PG is innervated by two neurons in each brain lobe; the release of the peptide, prothoracicotropic hormone (PTTH), by these neurons stimulates ecdysone production by the PG via the PTTH receptor Torso (Rewitz et al, 2009) and the Ras/MAPK signaling pathway (Caldwell et al, 2005). In *Drosophila* and in other insects such as *Manduca*, production of PTTH seems to be set in motion by achievement of a critical size or some correlate thereof (reviewed in Nijhout et al [2014], Boulan et al [2015]) and is dependent on circadian rhythms (Di Cara & King-Jones, 2016). Critical weight (CW) is defined as the minimum larval weight after which starvation no longer delays metamorphosis (Beadle et al, 1938; Mirth et al, 2005; Stieper et al, 2008). Indeed, once larvae are above CW (estimated to be approximately 0.8 mg in *Drosophila*), a mild acceleration to metamorphosis is often observed following starvation (Stieper et al, 2008). A related parameter is the minimum viable weight (MVW), which is operationally defined as the minimal weight required for 50% of larvae to survive to a particular developmental stage when starved (Mirth et al, 2005). The minimal viable weights needed for pupariation (MVW(P)) and eclosion (MVW

[1]Department of Molecular and Cell Biology, University of California, Berkeley, CA, USA [2]Department of Genetics, Cell Biology, and Development, University of Minnesota, Minneapolis, MN, USA

Correspondence: ikh@berkeley.edu

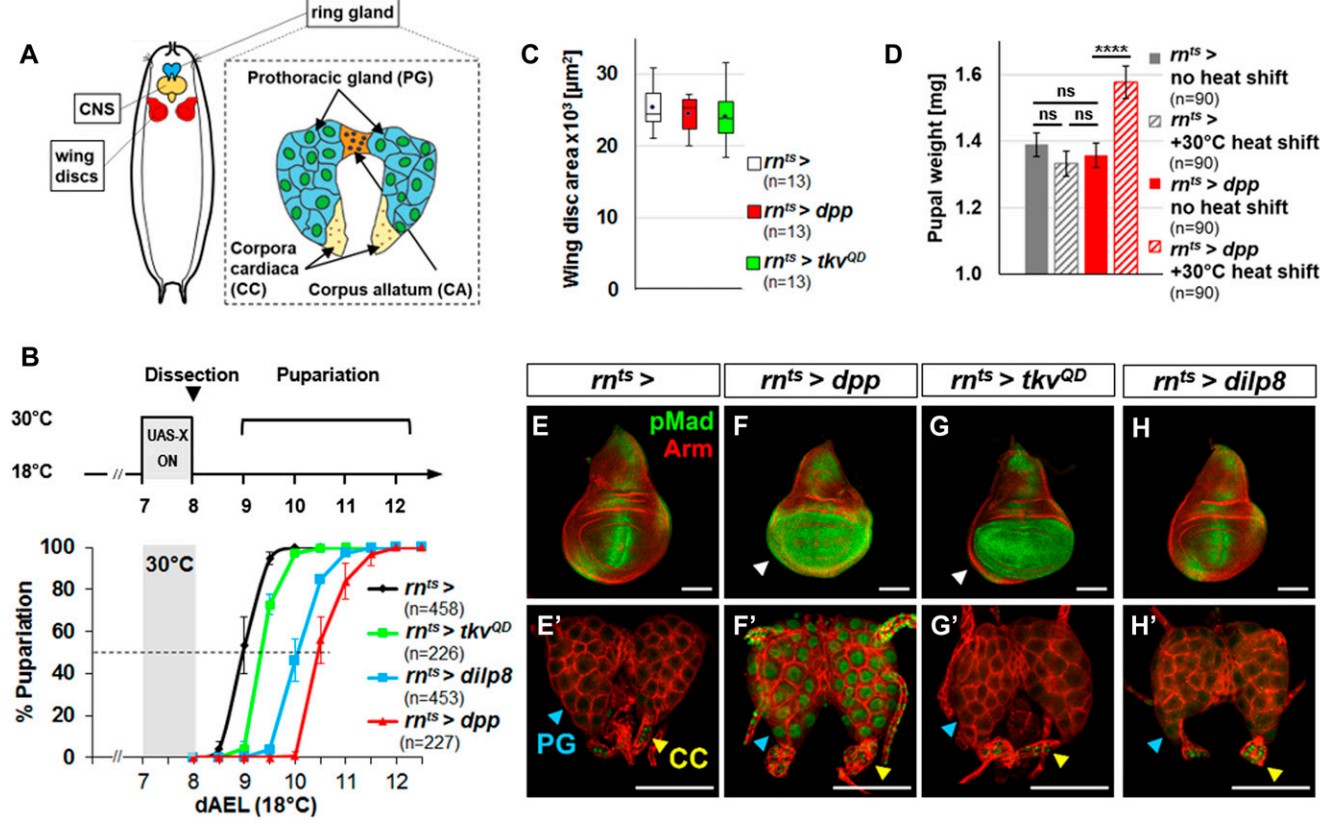

**Figure 1. Expression of Dpp in peripheral tissues induces Dpp signaling in the PG and delays pupariation.**
**(A)** Schematic of the larval ring gland. **(B)** Pupariation delay after a 24 h pulse of expression 7 d AEL induced by heat shift to 30°C: $rn^{ts}>dpp$ 36.5 ± 3.5 h; $rn^{ts}>tkv^{QD}$ 9.5 ± 3.0 h and $rn^{ts}>dilp8$ 26.0 ± 3.7 h. **(C)** Areas of wing discs dissected after 24 h of expression. **(D)** Pupal weights without heat shift: $rn^{ts}$ 1.39 ± 0.04 mg; $rn^{ts}>dpp$ 1.36 ± 0.04 mg. Pupal weights after 24 h of expression: $rn^{ts}>$ 1.33 ± 0.04 mg; $rn^{ts}>dpp$ 1.58 ± 0.05 mg. **(E–H)** Wing discs (E–H) and ring glands **(E'–H')** dissected after 24 h pulse of expression. Nuclear pMad is observed in the PG (blue arrowhead) in $rn^{ts}>dpp$, and in all genotypes in the CC (yellow arrowhead) and not in the CA. Data information: Error bars indicate standard deviations. ns, not significant; ***$P < 0.001$; ****$P < 0.0001$. Scale bars = 100 μm.

(E)) in *Drosophila* are approximately 0.68 mg and 0.98 mg, respectively (Stieper et al, 2008). The mechanism by which the size or weight of the *Drosophila* larva can impede or permit progression to the pupal stage is not known.

A major promoter of larval growth is the insulin signaling pathway. In parallel to its effects on other larval tissues, insulin receptor (InR)–mediated signaling in PG cells promotes ecdysone release (Colombani et al, 2005; Mirth et al, 2005). InR signaling may be necessary for the pulses of ecdysone production that occur during larval development and the large increase in ecdysone that drives the onset of metamorphosis. InR signaling promotes ecdysone production, in significant part, by cytoplasmic retention of the transcription factor FOXO (Koyama et al, 2014) and inhibition of the synthesis or action of the microRNA *bantam* in the cells of the PG (Boulan et al, 2013). This requirement of InR signaling for metamorphosis to occur appears necessary only before the attainment of CW. After that point, a reduction in InR activity no longer delays the timing of pupariation (Shingleton et al, 2005). This is consistent with a scenario where nutrient availability and the consequent growth need to reach a threshold value to drive this developmental transition. Conversely, under conditions of starvation, circulating Hedgehog made by enterocytes delays metamorphosis by inhibiting ecdysone production by the PG (Rodenfels

et al, 2014). Thus, the availability of food and its consequences can impact developmental timing in multiple ways. In addition, the Activin pathway also promotes the competence of the PG to respond to insulin and PTTH (Gibbens et al, 2011). Thus, multiple inputs seem capable of influencing developmental timing via a variety of diffusible signals that reach the PG. Within the PG, there is clearly crosstalk between these different pathways in ways that are not yet fully understood.

The growth status of imaginal discs, the larval primordia of adult structures such as wings and eyes, can also influence developmental timing. Although the complete absence of imaginal discs does not result in a change in developmental timing, growth abnormalities in imaginal discs can delay pupariation (reviewed in Jaszczak et al [2016]). Damaged or overgrown imaginal discs secrete the insulin/relaxin family member Dilp8 (Colombani et al, 2012; Garelli et al, 2012), which binds to receptors in the nervous system and PG (Colombani et al, 2015; Garelli et al, 2015; Vallejo et al, 2015; Jaszczak et al, 2016) and inhibits ecdysone production. Imaginal discs that grow slowly, as occurs in *Minute* mutants, also delay pupariation (Stieper et al, 2008; Parker & Shingleton, 2011). It is unclear whether this happens because of the growth status of discs or because disruptions in cell physiology elicited by *Minute* mutations activate a cellular stress response (Lee et al, 2018). A key unanswered question is whether

there is a mechanism that operates under normal physiological conditions to coordinate imaginal disc growth and maturation with entry into metamorphosis.

Here, we show that the morphogen Dpp, the *Drosophila* BMP2/4 ortholog, which has been studied extensively for its role in regulating growth and patterning within tissues, can also diffuse between tissues, such as between imaginal discs and the PG. Dpp signaling in the PG can negatively regulate ecdysone production, and moreover decreases as larvae approach metamorphosis. Our results suggest a role for Dpp as an inter-organ signal in the larva that regulates the timing of metamorphosis and has a role in the CW checkpoint.

# Results

### Dpp expressed in peripheral tissues can delay pupariation

The BMP2/4 ortholog, Dpp, functions as a morphogen to regulate growth and patterning within many tissues including imaginal discs (Hamaratoglu et al, 2014). We examined the effects of temporarily increasing *dpp* expression during the early third larval instar (L3) using *rn-Gal4* and a temperature-sensitive repressor, Gal80$^{ts}$ (hereafter *rn$^{ts}$>dpp*) (Fig 1B). *rn-Gal4* is expressed in wing discs and also some other tissues as assessed by the G-trace method (Evans et al, 2009) (Fig S1A). Under the conditions of this experiment, *rn$^{ts}$>dpp* did not increase wing disc size (Fig 1C) or adult wing size (not shown) but markedly delayed pupariation (Fig 1B) resulting in larger pupae, likely because of an extended growth phase (Fig 1D). Surprisingly, an activated form of the Dpp receptor Thickveins (Tkv$^{QD}$) (Nellen et al, 1996), (*rn$^{ts}$>tkv$^{QD}$*), which functions in cells autonomously and also does not affect disc size, elicited only a modest delay (Fig 1B and C). Consistent with the known ability of Dpp to spread within tissues, expression of *dpp*, but not *tkv$^{QD}$*, increased Dpp signaling beyond the wing pouch in the wing disc, as assessed by increased nuclear phosphorylated Mad (pMad) (Fig 1E–G, arrowheads in Fig 1F and G).

Because of the dramatic effects on pupariation timing caused by a relatively brief increase in *dpp* expression, we wondered whether Dpp could have effects on the ring gland. We therefore examined ring glands in these larvae for alterations in Dpp signaling. Within the ring gland, we observed increased levels of nuclear pMad in the PG with *rn$^{ts}$>dpp* but not *rn$^{ts}$>tkv$^{QD}$* (Fig 1E'–G'). Nuclear pMad was observed in the CC in all genotypes examined (Fig 1E'–G'). Because *rn-Gal4* is not expressed in the ring gland (Fig S1A), these observations suggest that Dpp might be reaching the ring gland from another location. One possibility is the central nervous system (CNS) because the ring gland is known to be innervated and because *rn-Gal4* is also expressed in the CNS (Fig S1A). However, when neuronal expression of *rn-Gal4* was prevented using *elav-Gal80*, there was no alleviation of the delayed pupariation, nuclear localization of pMad in the PG, or increased pupal size (Fig S1B–I). Thus, the Dpp that reaches the PG in these experiments is likely produced by more distant nonneuronal cells. Indeed, we found that when *dpp* was overexpressed in a variety of peripheral tissues using different driver lines, pupariation was always delayed suggesting that Dpp can reach the PG from a variety of locations in the larva (Fig S2A–G).

The insulin/relaxin-family member Dilp8 is produced by imaginal discs in response to injury, cell death, or various types of overgrowth. Although we did not observe disc overgrowth under the conditions of our experiment (Fig 1C), it is nevertheless possible that the presence of inappropriate amounts of Dpp could have promoted Dilp8 production, raising the possibility that Dpp delayed pupariation indirectly via Dilp8 production. *rn$^{ts}$>dpp* or *rn$^{ts}$>tkv$^{QD}$* caused little cell death and only a modest level of *dilp8* expression (Fig S3A–F). Moreover, *rn$^{ts}$>dpp* delayed pupariation even more than *rn$^{ts}$>dilp8* (Fig 1B) suggesting that *dpp* does not function upstream of *dilp8*. Conversely, *rn$^{ts}$>dilp8* did not affect nuclear pMad in the wing disc or PG (Fig 1H and H') indicating that Dilp8 does not activate Dpp signaling. Most importantly, *rn$^{ts}$>dpp* delayed pupariation in a *dilp8* mutant implying that Dpp can function independently of *dilp8* (Fig S3G). Taken together, these observations suggest that Dpp and Dilp8 induce delays in pupariation by separate pathways.

If Dpp reaching the ring gland is the cause of the delayed pupariation, then activating Dpp signaling autonomously in the ring gland should also delay pupariation. To test this hypothesis, we expressed *tkv$^{QD}$* using the ring gland driver *P0206-Gal4* (Colombani et al, 2005) and found that it delayed pupariation and increased pupal mass (Fig 2A–C). When we examined each component of the ring gland separately, pupariation was delayed using *phm-Gal4*, a PG-specific driver (Ono et al, 2006), but not using the *Aug21-Gal4* driver which is expressed in the CA (Adam et al, 2003) or the *AKH-Gal4* driver which is expressed in the CC (Fig 2D and E) (Lee & Park, 2004). All of these observations, taken together, point to the possibility that Dpp can reach the PG from peripheral tissues and that increased Dpp signaling in the PG can delay pupariation.

### Dpp expressed in peripheral tissues can reach the PG

To test more directly whether Dpp expressed in peripheral tissues could reach the PG, we used the *GFP-dpp* transgene that encodes a form of Dpp tagged with GFP which can activate Dpp signaling. When *GFP-dpp* (Entchev et al, 2000) was expressed using *rn-Gal4* (*rn$^{ts}$>GFP-dpp*), both GFP and nuclear pMad were detected in the PG but not in the immediately adjacent CA (Fig 3A and B) suggesting that its accumulation in the PG and the activation of the signaling pathway is dependent on binding to specific receptors. A different GFP-tagged secreted protein, atrial natriuretic factor (ANF-GFP) (Rao et al, 2001), expressed under the same conditions, was not detected in the PG (Fig 3C and D). Consistent with the possibility that Dpp can circulate in the hemolymph, a processed form of GFP-Dpp was detected in the hemolymph of *rn$^{ts}$>GFP-dpp* larvae (Fig 3E).

To determine whether Dpp can diffuse from discs to the PG in the absence of other tissues, we co-cultured *rn$^{ts}$>GFP-dpp* discs with wild type ring glands Fig 3F and G). Nuclear pMad was observed in the PG portion of those ring glands indicating that Dpp can diffuse from discs to the PG ex vivo. Nuclear pMad was not observed in the PG portion of ring glands cultured alone. Addition of hemolymph from *rn$^{ts}$>dpp* larvae was sufficient to elicit pMad nuclear localization (Fig 3H) indicating the presence of functional Dpp in the hemolymph of those larvae. Thus, Dpp can diffuse from discs to the hemolymph and from the hemolymph to the PG.

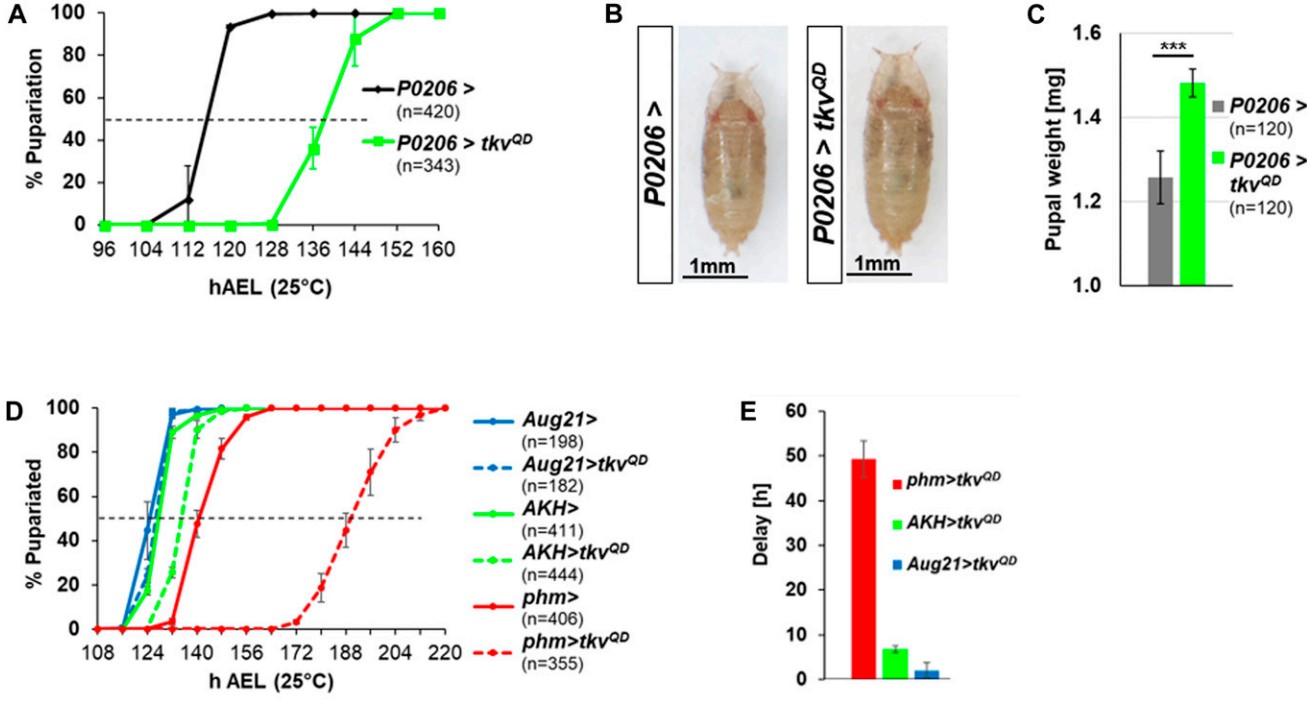

**Figure 2. Cell-autonomous Dpp signaling in the PG delays pupariation and increases organismal size.**
**(A)** *P0206>tkv^QD* has a 22.2 ± 2.1 h developmental delay. **(B)** Representative pupae of *P0206>* and *P0206>tkv^QD*. **(C)** Pupal weights are *P0206 >* 1.26 ± 0.06 mg; *P0206>tkv^QD* 1.48 ± 0.03 mg. **(D)** Effect of expressing *tkv^QD* in different parts of the ring gland on pupariation timing. *phm-Gal4* drives expression in the PG; *AKH-Gal4* is expressed in the CC; *Aug21-Gal4* is expressed in the CA. Mean pupariation times are: *phm >* 140.3 ± 1.5 h; *phm>tkv^QD* 189.7 ± 2.5 h; *Aug21 >* 124.3 ± 1.5 h; *Aug21>tkv^QD* 126.3 ± 0.6 h; *AKH >* 127.8 ± 0.3 h; and *AKH>tkv^QD* 134.7 ± 0.6 h. **(E)** Pupariation delays are shown in (D), normalized to the corresponding *Gal4* driver line alone: *phm>tkv^QD* 49.3 ± 4.0 h; *Aug21>tkv^QD* 2.0 ± 1.7 h; *AKH>tkv^QD* 6.8 ± 0.8 h. Data information: Error bars indicate standard deviations. ***P < 0.001. Scale bars = 1 mm.

If the Dpp receptor Tkv is indeed expressed in cells of the PG, then inactivating it in those cells should block the ability of Dpp in the hemolymph to activate the signaling pathway. To examine this possibility, we constitutively expressed *GFP-dpp* in discs using the lexOp-GAD system using *ap-GAD* and *lexOp-EGFP::dpp* (Fig 3I–M). Concurrently, we expressed *UAS-tkv^RNAi* in cells of the PG using *phm-Gal4*. Expression of *tkv^RNAi* in the PG-blocked Dpp signaling as assessed by the absence of nuclear pMAD staining. Thus, Tkv is necessary in cells of the PG to mediate the response of Dpp secreted by disc cells.

## Dpp signaling in the PG is developmentally regulated

Our experiments thus far indicate that Dpp expressed at increased levels in a variety of tissues including discs can diffuse to the PG and activate the signaling pathway via Tkv. What could be the physiological function of a pathway where Dpp from peripheral tissues acts on the PG to delay development? One possibility is that like Dilp8, Dpp is released from tissues such as imaginal discs under conditions of damage or overgrowth. To address this possibility, we induced apoptosis in discs either by targeted expression of the pro-apoptotic gene *eiger* (*egr*) or *reaper* (*rpr*) or with X-ray irradiation of larvae (Fig S4A–C and E–G). In all cases, we observed increased Dilp8 production in imaginal discs, but no increase in nuclear pMad in the PG. This was also the case with overgrown discs in *discs large* (*dlg*) hemizygotes (Woods & Bryant, 1991) (Fig S4D and H). Thus, Dpp signaling is not activated in the PG in response to tissue damage or

overgrowth and is therefore unlikely to function in a mechanism that slows development under those conditions.

To address a role for Dpp signaling in the PG during normal development, we extended our analysis to include ring glands at earlier time points during the final larval stage (L3) by visualizing nuclear pMad (Fig 4A–C) and two reporters, *dad-RFP* (Wartlick et al, 2011) (Fig 4D–I) and *brk-GFP* (Dunipace et al, 2013) (Fig 4D–F and J–L). *dad-RFP* expression increases with Dpp signaling, whereas *brk-GFP* expression increases as Dpp signaling decreases. In contrast to the PG in late L3 (120 h after egg lay [AEL]), we observed strong nuclear pMad in the PG in early L3 (72 h AEL). As larvae progress through L3, the levels of nuclear pMad and expression of *dad-RFP* decrease, whereas *brk-GFP* expression increases. Together, these observations indicate that Dpp signaling progressively decreases during the course of L3 in the PG. By comparison, Dpp signaling was consistently high in the CC (Fig 4A–C) and consistently low in the CA (Fig 4J–L). When Dpp signaling was reduced in the CC by overexpression of *dad*, pMad staining was reduced as well (Fig S5), indicating that it was unlikely to be a result of the anti-pMad antibody binding to another epitope. To our knowledge, the role of Dpp signaling in the CC is not known.

## Dpp signaling in the PG inhibits expression of ecdysone biosynthesis enzymes

The biosynthesis of ecdysone from cholesterol requires a number of enzymes encoded by genes known as the Halloween genes. In the

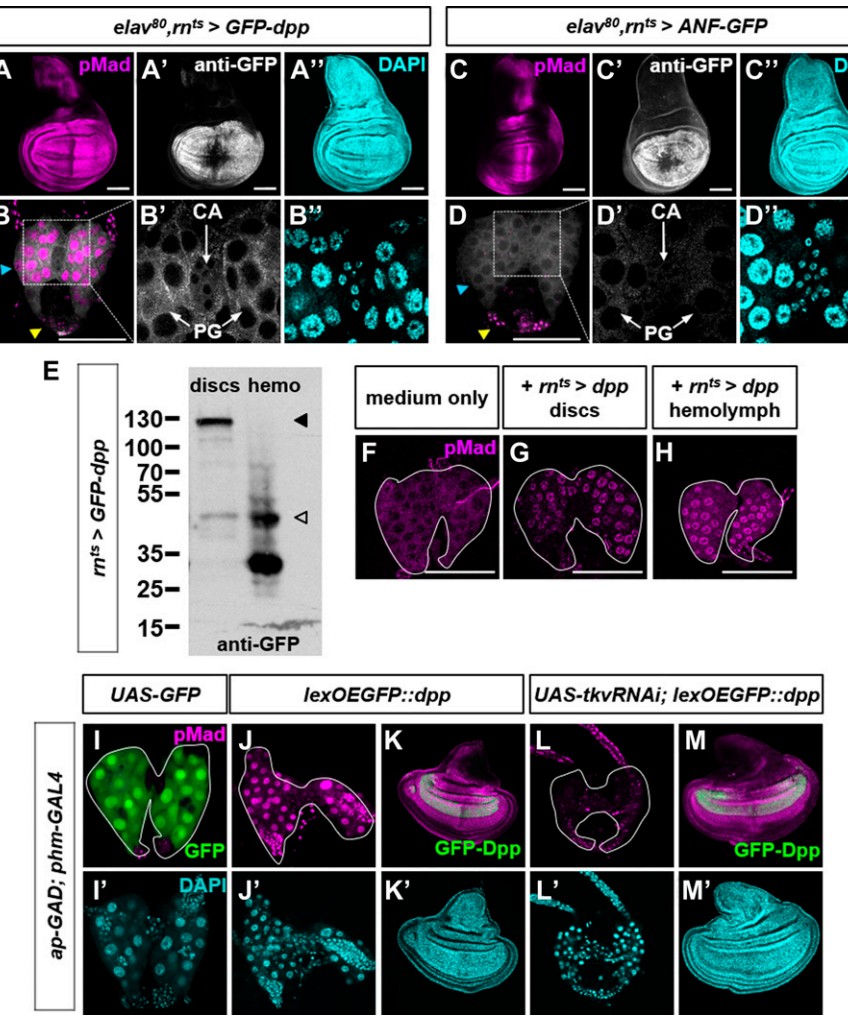

**Figure 3. Dpp can diffuse from peripheral tissues via the hemolymph to the PG and activate signaling via the Tkv receptor.**

**(A, B)** *elav-Gal80 rn^{ts}>GFP-dpp* wing disc **(A, A″)** and ring gland **(B, B″)**. Note GFP in PG but not CA **(B′)**. The blue arrowhead indicates PG and the yellow arrowhead indicates CC. **(C, D)** *elav-Gal80 rn^{ts}>ANF-GFP* wing disc **(C–C″)** and ring gland **(D–D″)**. **(E)** Western blot of *rn^{ts}>GFP-dpp* discs and hemolymph. Predicted unprocessed (filled arrowhead) and processed (unfilled arrowhead) forms. **(F–H)** Ex vivo cultured wild-type ring glands co-cultured with wing discs (G) and hemolymph (H) from *rn^{ts}>dpp* larvae. **(I–M)** Constitutive and simultaneous expression of *ap-GAD* in the dorsal wing disc and *phm-Gal4* in the PG. **(I, I′)** *UAS-GFP* confirms expression in the PG. *lexOEGFP::dpp* induces pMad activation in the PG **(J)** and local pMad and overgrowth in the wing disc **(K)**. However, no pMad activation occurs in the PG during simultaneous Tkv knockdown **(L)** despite the same local pMad and overgrowth in the wing disc **(M)** where Tkv is not knocked down. Data information: Scale bars = 100 μm.

wild-type PG, the levels of these enzymes increase toward the end of L3 as Dpp signaling decreases. Their up-regulation is known to occur, in significant part, at the level of transcription. To examine the consequence of changes in Dpp signaling in the PG, we examined the expression of several ecdysone biosynthesis enzymes (Ou & King-Jones, 2013) in FLP-out clones with altered Dpp signaling (Fig 5A). In early L3, when Dpp signaling is high, reducing Dpp signaling by overexpressing *dad* (Tsuneizumi et al, 1997) increased expression of Disembodied (Dib) (Fig 5B) or, to a lesser extent, Shadow (Sad) (Fig S6A) when compared with adjacent wild-type cells. In late L3, when cells in the PG have lower levels of Dpp signaling, augmenting Dpp signaling with activated Tkv reduced expression of Dib (Fig 5C), Shadow (Sad), Spookier (Spok), and Phantom (Phm) (Fig S6B–D). Thus, Dpp signaling in the PG negatively regulates the expression of multiple ecdysone biosynthesis enzymes. As a consequence, when Dpp signaling decreases in late L3, this would be expected to alleviate the repression of Halloween genes and allow an increase in ecdysone biosynthesis and hence entry into metamorphosis.

These findings suggest that the mechanism by which artificially increasing Dpp signaling in the PG delays pupariation is, at least in part, by reducing the levels of enzymes required for ecdysone biosynthesis. If so, then providing exogenous ecdysone might overcome this limitation. Feeding larvae the active form of ecdysone, 20-Hydroxyecdysone (20E) reduced the delay in pupariation caused by increased Dpp signaling in the PG (Fig 5D) and also reduced the delay-induced increase in pupal mass (Fig 5E).

## Reducing Dpp signaling in the PG has minor effects under fed conditions but abrogates the critical-weight checkpoint under conditions of starvation

Because reducing Dpp signaling in the PG in early L3 increased expression of at least a subset of Halloween genes (Figs 5B and S6A), it could potentially also result in early pupariation. We have previously shown that expression of *UAS-tkv^{RNAi}* in the PG using *phm-Gal4* can prevent the increase in Dpp signaling elicited by overexpression of Dpp in discs (Fig 3L). Thus, at endogenous levels of Dpp expression, *phm>tkv^{RNAi}* larvae would be expected to have reduced levels of Dpp signaling in the PG. We tested different *tkv^{RNAi}* and *Mad^{RNAi}* lines, a dominant-negative form of *tkv* (*tkv^{D95K}*), or *dad* expressed using *phm-Gal4* (Fig S7A–C). In all cases, the larvae developed normally through all three instars, and we did not

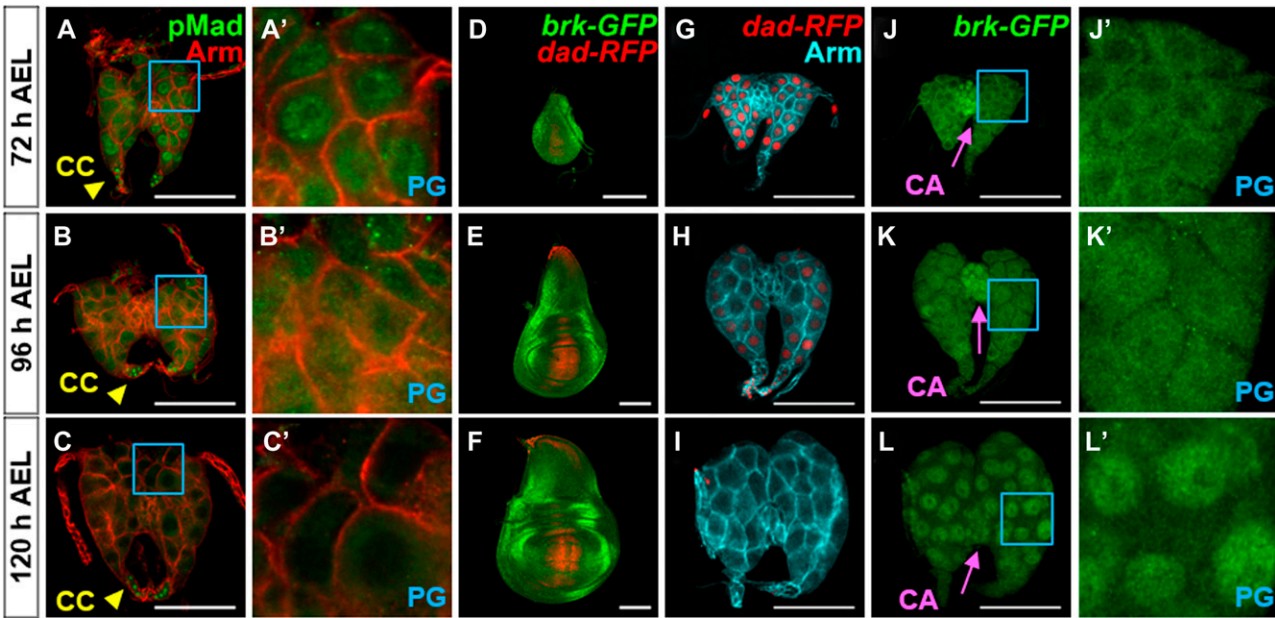

**Figure 4. Dpp signaling in the PG decreases during L3.**
**(A–C)** Ring glands at 72, 96, and 120 h AEL at 25°C. **(A'–C')** Blue boxed region showing PG cells at higher magnification. **(D–F)** Wing discs expressing *dad-RFP* and *brk-GFP*. **(G–I)** Ring glands expressing *dad-RFP*. **(J–L)** Same ring glands as in (G–I) expressing *brk-GFP*. **(J'–L')** Blue boxed regions showing PG cells at higher magnification. Data information: Scale bars = 100 μm.

observe a robust and reproducible acceleration of pupariation or a reduction in pupal size that would be expected to be a consequence of premature pupariation. A subset of the lines tested elicited a modest (2–3 h) acceleration. However, given the variability of genetic backgrounds and the comparatively slow developmental timing of the *phm-Gal4* driver line on its own (Fig 2D and legend), it is difficult to evaluate the relevance of these small effects. Reducing the availability of Dpp ligand by *dpp^RNAi* expression using *dpp-Gal4* also did not induce precocious pupariation (Fig S7D). Over-expression of *brinker* (*brk*) in the PG, which would be expected to repress many Dpp target genes, arrested larvae in L2, yet caused wandering (prepupal) behavior (Fig S7E–Q). Although this observation is difficult to interpret, it is similar to the precocious metamorphosis observed with increased Activin signaling (Gibbens et al, 2011). Although we do not currently understand the mechanism underlying this L2 arrest, it may further illustrate how the Dpp and Activin signaling pathways act in opposite ways. Another possibility for this L2 arrest is that the overexpression of *brk* at this early stage may simply be toxic to the PG. Because of the small size of the ring gland at this early stage, we were unable to dissect and examine the PG for cell death. Taken together, our experiments do not provide convincing evidence that reducing Dpp signaling in the PG can accelerate the onset of pupariation and therefore suggest that reducing Dpp signaling in the PG may, in itself, be insufficient to trigger the onset of metamorphosis. However, because maintaining elevated levels of Dpp signaling in the PG delays pupariation, a reduction in Dpp signaling in late L3 is necessary, in combination with other signals, for timely metamorphosis.

Although disruption of Dpp signaling in the PG did not elicit a robust effect on developmental timing under fed conditions, we wondered if it had a role in mediating the CW or MVW checkpoints,

both of which in *Drosophila* occur approximately 8 h after L3 ecdysis (AL3E). When larvae were starved beginning at 2 h AL3E, larvae carrying either the *phm-Gal4* or *UAS-Mad^RNAi* transgene did not pupariate and instead arrested development as larvae (Fig 5F, G, and K). When the larvae were transferred to starvation conditions beginning at 4 h or 6 h AL3E, a pupariation delay was observed. When starvation was initiated after the 8 h AL3E, no delay in developmental timing was observed. This is consistent with the time when CW is reached. Strikingly, when *phm>Mad^RNAi* larvae were moved to starvation conditions at 2 h AL3E, 72% were able to form pupae (Fig 5H) which were much smaller than *phm>Mad^RNAi* pupae obtained under fed conditions (Fig 5L). An even greater percentage of larvae pupariated when starvation was commenced at 4 h or 6 h AL3E (Fig 5I). When starvation was commenced at 8 h AL3E, no developmental delay was observed in all genotypes. Most of the small *phm>Mad^RNAi* pupae generated when starvation was commenced at 0–6 h AL3E did not develop to viable adults (Fig 5J). The small, subviable pupae observed under these conditions are similar to those observed when ecdysone is fed to larvae shortly after the L2/L3 ecdysis (Koyama et al, 2014). Thus, although reducing the level of Dpp signaling in the PG has only subtle effects when food is abundant, under conditions of starvation, it appears to abrogate a mechanism that functions to prevent larvae that have grown insufficiently from proceeding to pupariation.

### Dpp signaling in the PG in early L3 requires Dpp expression in imaginal discs

To help identify the source of Dpp that reaches the PG, we expressed *UAS-dpp^RNAi* using Gal4 drivers and examined pMad levels in the PG at 72 h AEL when nuclear pMad is normally

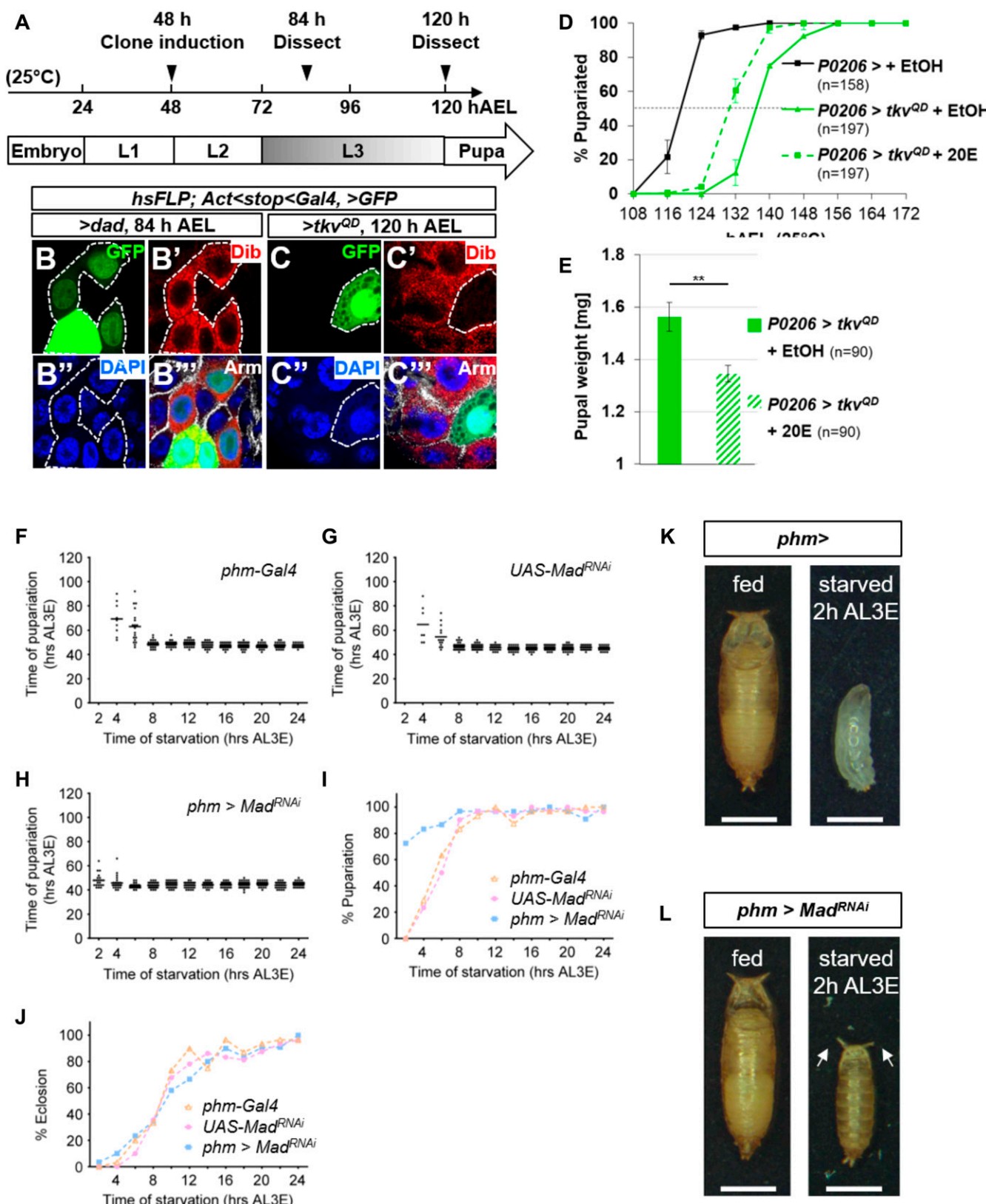

**Figure 5. Dpp signaling in early L3 represses ecdysone biosynthesis enzymes and prevents precocious pupariation during starvation.**
**(A)** Schematic of heat-shock clone induction experiments. Cultures were maintained at 25°C. FLP-out clones (*act<stop<Gal4*) were induced 48 h AEL. FLP-out clones in the PG show uneven expression of *UAS-GFP* possibly because of asynchrony of or variability in the number of endocycles between the individual polyploid cells of the PG. **(B, C)**

observed. The *dpp-Gal4* driver is expressed only in a subset of cells that express Dpp. Nevertheless, in *dpp-Gal4, UAS-dpp^RNAi* larvae, the level of nuclear pMad in the PG is greatly reduced (Fig 6A and B). Thus, the Dpp that reaches the PG must be produced by cells with current or past expression of *dpp-Gal4*. In addition, the ability of *dpp-Gal4, UAS-dpp^RNAi* to reduce pMad levels in the PG indicates that the pathway is indeed mostly being activated by Dpp and not by other ligands capable of activating the same signaling pathway such as Gbb (Haerry et al, 1998).

We used G-TRACE (Evans et al, 2009) to identify both current (RFP) and previous (GFP) expression of *dpp-Gal4* (Fig 6C–H). Expression was strongest in in imaginal discs (Figs 6C and Fig S8A–C), the salivary glands (Fig 6D), and some portions of the CNS (Fig 6E). Expression was not observed in the ring gland (Fig 6F), gut (Fig 6G), fat body (Fig 6H), or lymph gland (Fig S8D). Others have reported expression of a *dpp-Gal4* transgene in scattered cells in the gut of late L3 larvae (Denton et al, 2018) and therefore lower levels of expression may escape detection by this method. A previous study (Huang et al, 2011) reported the expression of a *dpp-lacZ* reporter in the CA of the ring gland. In that same study, *dpp* expression in the ring gland, measured using quantitative RT-PCR from total RNA prepared from ring glands, was reported to increase in late L3, peaking in wandering L3 larvae. We took advantage of stocks constructed recently where gene editing has been used to tag endogenous Dpp with an HA-tag (Bosch et al, 2017). Using anti-HA antibody, we detected Dpp in the stripe of Dpp producing cells in the wing disc and in cells of the CC even in late L3 ring glands (when no pMad is observed in the PG), but no expression in the PG or CA (Fig 6I and J). Furthermore, we could not detect expression of a *lacZ* reporter of *dpp* in the CA using three *different dpp-lacZ* lines obtained from the Bloomington *Drosophila* Stock Center (data not shown). Moreover, when we expressed *UAS-GFP-dpp* in the CC using the *AKH-Gal4* driver, we did not observe any pMad in the PG (Fig 6K and L) indicating that Dpp is unlikely to reach the PG from the CC. Expression of *UAS-GFP-dpp* in the CA using *Aug21-Gal4* resulted in pMad being observed only in PG cells immediately adjacent to the CA; uniform nuclear pMad throughout the PG was not observed (Fig 6M and N). These observations indicate that other portions of the ring gland are unlikely to produce the Dpp that reaches the PG. Importantly, because *dpp-Gal4* is not expressed in the ring gland and *dpp-Gal4, UAS-dpp^RNAi* can reduce pMad in the PG in early L3 larvae, the Dpp must mostly originate in tissues other than the ring gland.

In the CNS, *dpp-Gal4* is expressed in the optic lobes and parts of the ventral nerve cord (Fig 6E) and the PTTH-expressing neurons which innervate the PG (Fig S8E and E'). When using *dpp-Gal4* and *UAS-tdTom*, tdTom fluorescence can be visualized in the cell bodies

of the PTTH neurons and the axonal branches that innervate the PG (Fig S8F) that show punctate staining with anti-PTTH (Fig S8F'). In the presence of *elav-Gal80*, these branches still show staining with anti-PTTH, but no longer have tdTom fluorescence (Fig S8G and G') confirming the neuronal characteristics of these cells. However, expression of *GFP-dpp* in PTTH-producing neurons does not reach the PG nor is nuclear pMad observed in PG cells when *GFP-dpp* is expressed using *ptth-Gal4* (Fig S8H–K). In addition, expression of *dpp^RNAi* either using *elav-Gal4* (which is expressed in most neurons) (Fig S8L, M, and P) or *ptth-Gal4* does not reduce pMad expression in the PG (Fig S8N and O), indicating that the Dpp is unlikely to be from the PTTH neurons or other neurons.

Other than the CNS, *dpp-Gal4* is expressed in the salivary glands (Fig 6D) and in the imaginal discs (Figs 6C and S8A–C). However, *dpp* itself is not expressed in the salivary glands either in embryos (St Johnston & Gelbart, 1987) or in wandering L3 larvae (Brown et al, 2014). Taken together, these experiments suggest that the Dpp that reaches the PG is mostly from the imaginal discs. In support of this possibility is recent work that shows that Dpp is secreted basolaterally in imaginal discs, that it can reach the basement membrane, and, at least under conditions of overexpression, can reach distant pericardial cells (Harmansa et al, 2017; Ma et al, 2017). Our experiments, however, cannot exclude the possibility that Dpp is secreted into the hemolymph in a stage-dependent manner by a small population of cells that has escaped detection using our methods.

## Interaction of Dpp signaling with pathways that regulate ecdysone production in the PG

Several signaling pathways have been shown to regulate ecdysone production by the PG, most notably the insulin-PI3K pathway (Colombani et al, 2005; Mirth et al, 2005), the Ras-MAPK pathway (Caldwell et al, 2005), the *bantam* microRNA (Boulan et al, 2013) and the Activin pathway (Gibbens et al, 2011). Each of these pathways can be manipulated to delay pupariation. To test whether these pathways delay pupariation by increasing Dpp signaling in the PG, we examined late L3 ring glands where either the insulin-PI3K, Ras, or Activin pathways were inhibited or where *ban* was overexpressed for alterations in pMad levels. None of these manipulations caused an increase in pMad in the PG indicating that these pathways do not delay pupariation by augmenting Dpp signaling (Fig 7A–F).

We then tested whether Dpp signaling could function upstream of any of these pathways. Signaling via the insulin-PI3K pathway results in the phosphorylation and cytoplasmic retention of the FOXO transcription factor. Activation of Dpp signaling in the PG resulted in increased levels of nuclear FOXO indicating that Dpp

FLP-out clones induced 48 h AEL, ring glands dissected 84 h AEL or 120 h AEL, examined for Disembodied (Dib) expression. **(B–B''')** GFP-expressing cells expressing the Dpp signal inhibitor Dad have enhanced Dib expression. **(C–C''')** GFP-expressing cells expressing the Dpp signal activator *UAS-tkv^QD* have lower Dib expression. **(D, E)** Effect of feeding 20-hydroxyecdysone (20E) to *P0206>tkv^QD* larvae. Mean pupariation times: *P0206>* +EtOH 119.0 ± 0.5 h; *P0206>tkv^QD* + EtOH 136.8 ± 0.3 h; *P0206>tkv^QD* 128.8 ± 2.9 h (D) and pupal weight (E). **(F–H)** Pupariation times of larvae that were initially raised on food and then starved beginning at different 2 h-interval time points after L3 ecdysis. Both *phm-Gal4* (F) and *UAS-Mad^RNAi* (G) larvae are not viable when starvation begins at 2 h AL3E and pupariate only after a delay when starved beginning at 4 h AL3E (median pupariation time is 68 h AL3E for *phm-Gal4* and 56 h AL3E for *UAS-Mad^RNAi*) or 6 h AL3E (median pupariation time is 62 h AL3E for *phm-Gal4* and 52 h AL3E for *UAS-Mad^RNAi*). Beginning at 8 h AL3E, starvation no longer induces a pupariation delay and the median pupariation times are between 47.5 h and 50 h AL3E for *phm-Gal4* and between 44 h and 46 h AL3E for *UAS-Mad^RNAi*. *phm>Mad^RNAi* (H) larvae are viable and able to pupariate when starvation begins at 2 h AL3E. At this and all following starvation time points, there is no delay and median pupariation times are between 44 h and 48 h AL3E (n = 28–32 for each time point). **(I)** Percentages of larval populations that pupariate (n = 28–32 for each time point). **(J)** Percentages of adult eclosion (n = 28–32 for each time point). **(K)** *phm-Gal4* larvae pupariate as wandering L3 larvae when continuously raised on food and do not pupariate when starved at 2 h AL3E. **(L)** *phm>Mad^RNAi* larvae pupariate as wandering L3 larvae when continuously raised on food and 72% of larvae form small early L3 pupae with everted spiracles (arrows) when starved at 2 h AL3E. Data information: **P < 0.01. Scale bars are 1 mm.

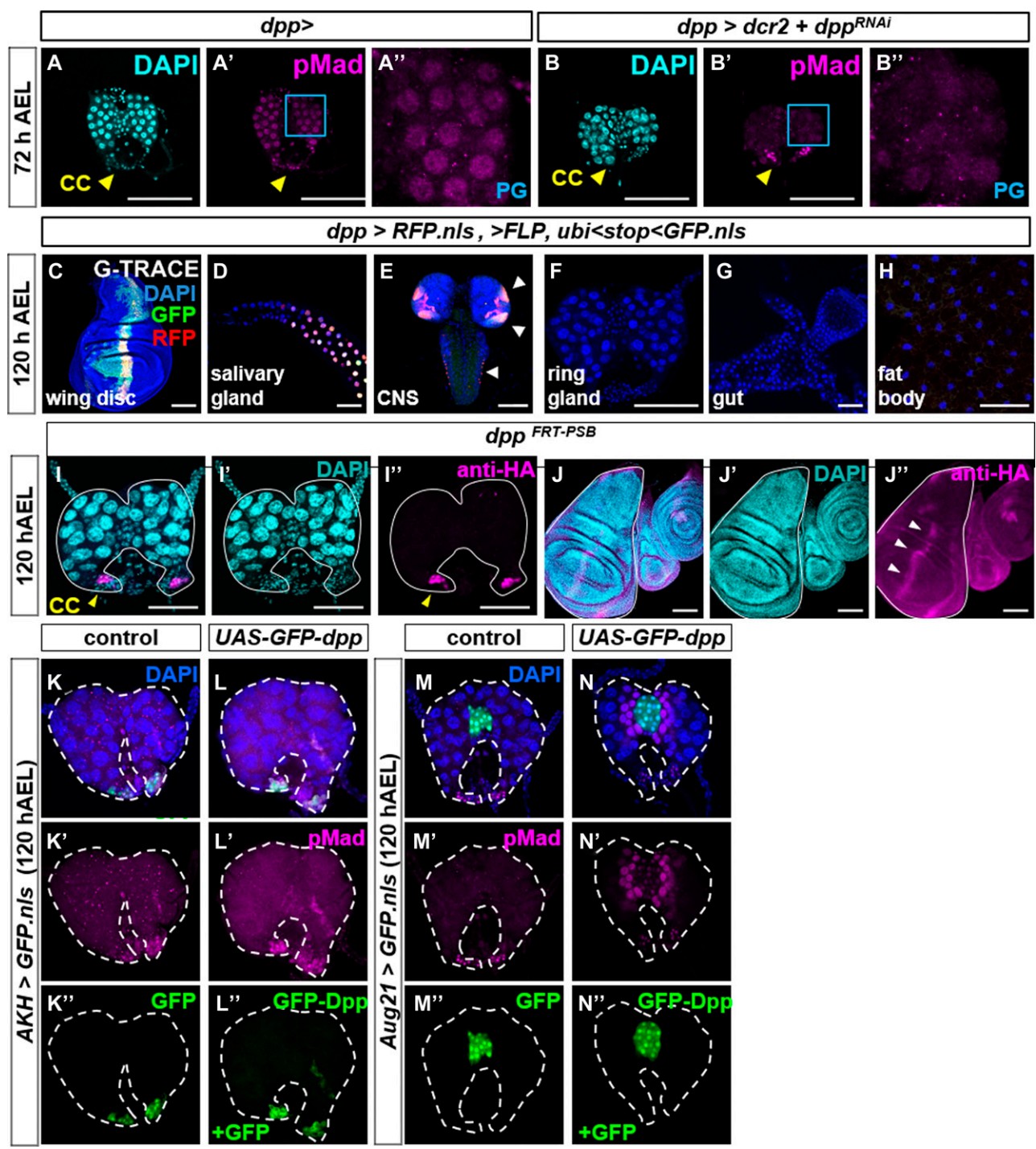

**Figure 6.  The PG receives Dpp from *dpp-Gal4* expressing cells and not from the CA or CC.**
**(A, B)** Effect of *dpp-Gal4, UAS-dpp^RNAi^* in ring glands dissected at 72 h AEL. Nuclear pMad is detected in PG cells of control *dpp-Gal4* **(A–A")** but not in PG cells of *dpp-Gal4, UAS-dcr2, ± UAS-dpp^RNAi^* **(B–B")**. **(C–H)** *dpp-Gal4* expression analyzed using G-TRACE. Current (RFP) or prior (GFP) expression. **(I, J)** Expression of HA-tagged Dpp protein expressed by the allele *dpp^FRT-PSB^*. HA-tagged Dpp is expressed in CC cells, indicated by yellow arrowhead (I) and in the anterior stripe of wing discs, indicated by white arrowheads (J). **(K, L)** Ring glands in which *UAS-GFP* **(K–K")** or *UAS-GFP + UAS-GFP-dpp* **(L–L")** is overexpressed in CC cells. **(M, N)** Ring glands in which *UAS-GFP* **(M–M")** or *UAS-GFP + UAS-GFP-dpp* **(N–N")** is overexpressed in CA cells.

signaling can act antagonistically to the insulin-PI3K pathway in regulating FOXO localization (Fig 7G–I). It has been shown that nuclear FOXO acts indirectly to increase the level of the *ban*

microRNA (Boulan et al, 2013). Consistent with this observation, expression of *UAS-tkv^QD^* in the PG increases *ban* levels as assessed by reduced expression of the *ban* sensor (Fig 7J). The antagonism

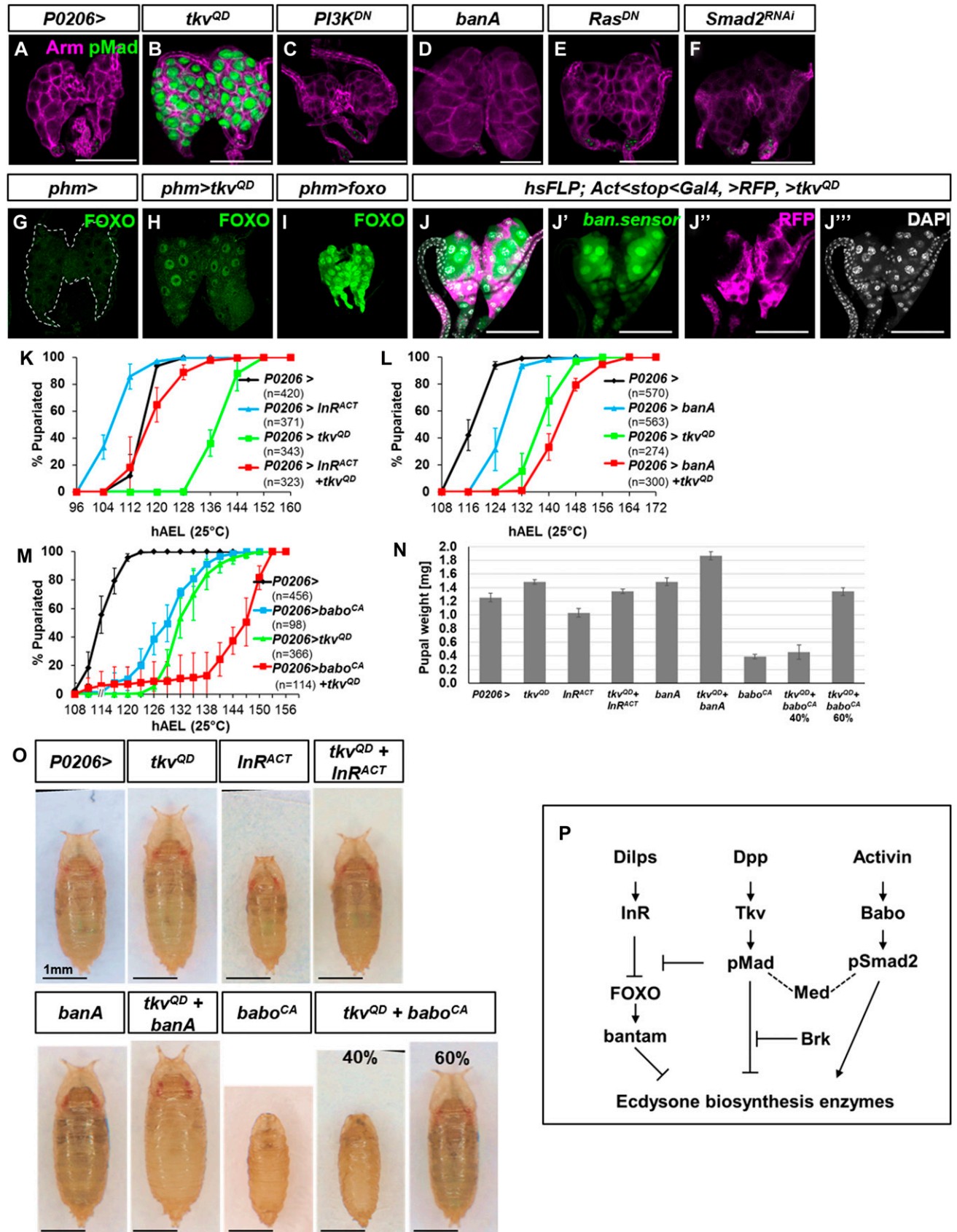

between the insulin-PI3K and Dpp pathways in the PG is also evident from their opposite effects, when activated, on the timing of pupariation and pupal size (Fig 7K, N, and O). In contrast, both increasing Dpp signaling and *ban* overexpression delay pupariation (Fig 7L, N, and O). Because increased Dpp signaling elicits a much greater pupariation delay than *ban* overexpression, and because simultaneous activation of both pathways causes a greater delay than activation of either alone, it is likely that each also has effects on the timing of pupariation that are independent of the other.

The Activin pathway appears to function antagonistically to the Dpp pathway in the regulation of imaginal disc growth (Peterson & O'Connor, 2013) although the mechanism by which the two pathways interact has not been elucidated. The interaction of these two pathways in the PG also seems complicated. Although the expression of an activated form of the Activin receptor Baboon (Babo$^{CA}$) elicits a similar delay to the expression of an activated form the Dpp receptor Tkv (Fig 7M), the appearance of the pupae is quite different. Activation of Tkv in the PG cells results in an extended L3 stage and larger pupae. However, with Babo$^{CA}$, larvae have an extended L2 stage before pupariating stage precociously and giving rise to smaller pupae (Fig 7N and O) that fail to evert their spiracles (Gibbens et al, 2011). This spiracle eversion phenotype is observed in other mutants where ecdysone levels are high as occurs in *ban* mutants consistent with antagonistic effects of these two pathways on ecdysone biosynthesis (Boulan et al, 2013). Concurrent activation of both pathways results in an even stronger delay in pupariation. However, when the pupae are examined, they seem to fall into two groups—approximately 60% of the pupae resemble Tkv$^{QD}$-expressing pupae, whereas 40% resemble Babo$^{CA}$-expressing pupae. Thus, as in the wing disc (Peterson & O'Connor, 2013), there does not seem to be a simple mechanism which accounts for the interaction between these pathways. The interactions of Dpp signaling with pathways previously implicated in regulating ecdysone production by the PG are summarized in Fig 7P.

## Discussion

The morphogen Dpp has mostly been studied for its roles in regulating growth and patterning within tissues where a subset of cells secrete Dpp and the remaining cells respond to Dpp. Here, we provide evidence that Dpp expressed in a variety of tissues can reach distant tissues via the hemolymph and impact the physiology of those tissues. Although we have focused on the effects in the PG, it is possible, even likely, that circulating Dpp can have effects on many tissues and can therefore also function like a hormone.

### Dpp as an inter-organ signal

How could Dpp secreted by discs reach the hemolymph? Although it has been reported previously that Dpp is secreted from the disc proper epithelium into the lumenal space enclosed by the peripodial epithelium (Gibson et al, 2002), recent work using nanobodies that can detect GFP-tagged Dpp proteins suggests that Dpp is mostly secreted basolaterally and that it can be detected within the extracellular matrix of the basement membrane (Harmansa et al, 2017). Although our experiments show the presence of overexpressed, tagged versions of Dpp in the hemolymph, the detection of native Dpp has been challenging in the study of *Drosophila* morphogens and has only been accomplished by few (Panganiban et al, 1990; Akiyama & Gibson, 2015). We could not detect physiological levels of Dpp in the hemolymph, either using antibodies to detect native Dpp or anti-HA antibodies to detect endogenously tagged Dpp, mainly because of the limited volume of hemolymph that can be extracted manually while avoiding clotting and because of the high background produced on Western blots. Notably, others have detected Dpp that is overexpressed in discs in pericardial cells which filter the hemolymph (Ma et al, 2017). Based on these recent findings, it would be expected that a small fraction of Dpp expressed in discs at endogenous levels would indeed reach the hemolymph.

**Figure 7. Dpp signaling acts upstream of *bantam* and FOXO to regulate ecdysone synthesis.**
**(A–F)** Ring glands dissected at 120 h AEL. pMad is not activated by other pathways known to inhibit ecdysone synthesis. (A) *P0206>*; (B) *P0206>tkv$^{QD}$*; (C) *P0206>PI3K$^{DN}$*; (D) *P0206>banA*; (E) *P0206>Ras$^{DN}$*; (F) *P0206>Smad2$^{RNAi}$*. Other than *P0206>tkv$^{QD}$*, none of these transgenes that have each been reported to cause a delay in pupariation, cause nuclear pMad accumulation in the PG, indicating that these pathways do not activate Dpp signaling in the PG. **(G–I)** FOXO protein visualized in ring glands using anti-FOXO antibody. FOXO is cytoplasmic in *phm>* ring glands (G) and nuclear in *phm>FOXO* ring glands (I). However, in 20% of *phm>tkv$^{QD}$* ring glands, we observed nuclear FOXO (H). The reason for the incomplete penetrance of this phenomenon is unclear. A similar frequency of nuclear FOXO was observed when using two different transgenes that encode activated *tkv* and also when using the *P0206-Gal4* instead of *phm-Gal4*. Thus, Dpp signaling could impact the insulin pathway upstream of FOXO. **(J)** Ring glands with FLP-out clones expressing *UAS-tkv$^{QD}$* and RFP. Reduced *ban* sensor expression (green in **J, J'**) indicates increased *ban* expression. **(K–O)** Activation of different signaling pathways in the ring gland using *P0206-Gal4*. *phm-Gal4* was not used because that driver line itself has a developmental delay. **(K)** Interaction of Dpp signaling with the insulin pathway in the PG. *UAS-tkv$^{QD}$* delays pupariation. Expression of an activated form of the insulin receptor (*UAS-InR$^{ACT}$*) accelerates pupariation. With concurrent expression of both transgenes, pupariation timing is similar to that of the *P0206-Gal4* driver alone. Pupariation times were *P0206 > 115.8 ± 0.3 h*; *P0206>tkv$^{QD}$ 140.7 ± 7.2 h*; *P0206>InR$^{ACT}$ 106.3 ± 1.3 h*; and *P0206>tkv$^{QD}$ + InR$^{ACT}$ 117.0.7 ± 3.0 h*. Similarly, *P0206> InR$^{ACT}$* pupae are small, *P0206>tkv$^{QD}$* pupae are large, and *P0206> InR$^{ACT}$ + tkv$^{QD}$* pupae are similar in size or weight to *P0206>* pupae (N, O). **(L)** Interaction of Dpp signaling with *ban*. *UAS-banA* also delays pupariation. Pupariation times were *P0206 > 117.0 ± 1.0 h*; *P0206>tkv$^{QD}$ 138.0 ± 2.6 h*; *P0206>banA 125.7 ± 1.5 h*; and *P0206>tkv$^{QD}$ + banA 143.0 ± 1.7 h*. Thus, co-expression of *UAS-tkv$^{QD}$* and *UAS-banA* delays pupariation more than either transgene alone. Because the delay elicited by *UAS-banA* overexpression is less than that obtained with *UAS-tkv$^{QD}$*, it is unlikely that *tkv$^{QD}$* delays pupariation exclusively by increasing *ban* expression. Similarly, the increase in pupal size or weight obtained with co-expression of *UAS-tkv$^{QD}$* and *UAS-ban* is greater than the effect obtained with either transgene alone (N, O). Note that even though the pupariation delay obtained with *UAS-banA* is less than that obtained with *UAS-tkv$^{QD}$*, the effects on pupal size are similar suggesting that *UAS-banA* is more effective than *UAS-tkv$^{QD}$* in increasing growth rate. **(M)** Interaction of Dpp signaling with Activin signaling. Pupariation times were *P0206 > 114.5 ± 1.3 h*; *P0206>tkv$^{QD}$ 129.2 ± 1.5 h*; *P0206>babo$^{CA}$ 133.5 ± 3.1 h*; and *P0206>tkv$^{QD}$ + babo$^{CA}$ 148.0 ± 3.0 h*. Expression of a constitutively active form of the Activin receptor (*UAS-babo$^{CA}$*) delayed development. However, the larvae appeared delayed in L2, generating small pupae as has been described previously. With co-expression of *tkv$^{QD}$* and *babo$^{CA}$*, 40% of the pupae resembled P0206>*babo$^{CA}$* pupae in appearance and the remaining 60% resembled pupae that had gone through the L3 stage. The delay in pupariation was close to being additive of the delays caused by either *P0206>tkv$^{QD}$* or *P0206>babo$^{CA}$* alone. **(N)** Pupal weights of all of the conditions shown in panels (A–C). Summary of tests of statistical significance of pairwise comparisons of pupal weight measurements is in Table 1. **(O)** Images of pupae of all the conditions shown in panels (K–M). **(P)** Model for interaction of Dpp pathway with the InR/FOXO/*bantam* and β-Activin pathways. Data information: Scale bars = 100 μm, except in (O) where they are 1 mm.

Our experiments also demonstrate that the Dpp receptor Tkv is expressed on cells of the PG and that it is capable of responding to circulating Dpp. We have shown that the overexpression of *dpp* using *ap-LexGAD* induces pMad activation in the PG and that this effect can be suppressed by expressing *tkv*[RNAi] in the cells of the PG. Taken together, our experiments demonstrate that Dpp is capable of functioning as an inter-organ signal.

### Dpp signaling in the PG regulates ecdysone production

By manipulating Dpp signaling in subsets of cells in the PG using genetic mosaics, we have shown that Dpp signaling negatively regulates ecdysone production, at least in part, by regulating the levels of enzymes required for ecdysone biosynthesis. The InR signaling pathway has also been shown to regulate the levels of these same enzymes, and this action of InR signaling seems to occur by down-regulating levels of the *bantam* microRNA. Because increasing Dpp signaling can result in increased nuclear localization of FOXO and an increase in *ban* activity, Dpp signaling must modulate InR signaling upstream of FOXO. Our observations do not preclude additional levels of crosstalk. The Activin pathway has been shown to function antagonistically to Dpp signaling in regulating disc growth by an uncharacterized mechanism (Peterson & O'Connor, 2013). A similar antagonism between the two pathways also appears to function in the PG. It can be speculated that this antagonism may occur at the level of Medea, a co-Smad that can associate with either pMad or pSmad2 to activate either the Dpp or Activin pathway downstream. If the initial activation of one of these pathways then inhibits the activation of the other, this could explain why we sometimes observe a developmental delay (Dpp) or a developmental arrest (Activin) under conditions where we activate both pathways simultaneously in the ring gland. A better mechanistic understanding of the interaction of these pathways could explain how inputs from different tissues are integrated in the PG to regulate ecdysone production. In addition, in butterflies, a TGF-$\beta$ ligand and a Dpp-type ligand have both been shown to regulate JH by the CA (Ishimaru et al, 2016). Thus far, we have not observed binding of Dpp-GFP to the CA (Fig 3B); other ligands could potentially have similar functions in *Drosophila*.

### Antagonizing Mad function in the PG abrogates the critical-weight checkpoint

Although reducing Dpp signaling in well-fed larvae had modest effects on developmental timing, expression of a *UAS-Mad*[RNAi] transgene in the PG abrogated the critical-weight checkpoint. When these larvae were starved shortly after the L2/L3 ecdysis, rather than arresting as larvae, they proceeded to generate small pupae that did not develop into viable adults likely because they had not accumulated enough stored nutrients to sustain metamorphosis. A possible explanation is that starvation might normally delay the decline in Dpp signaling in the PG which, based on our findings, would be expected to delay pupariation. By antagonizing Dpp signaling in the PG, the mechanism that would normally delay pupariation would be subverted and allow larvae to proceed to pupariation even if they have not grown sufficiently.

### Why does Dpp signaling decrease in the PG during L3?

Our data show that under normal physiological conditions, Dpp signaling in the PG decreases as Halloween gene expression increases. This is consistent with the observation that Dpp signaling negatively regulates ecdysone production. As larvae mature, reduced Dpp signaling in the PG would be expected to allow ecdysone biosynthesis and thereby promote the onset of metamorphosis. This finding, however, seems at odds with the observation that Dpp levels, at least in wing discs, increase as discs grow during L3 (Wartlick et al, 2011). If Dpp from discs can pass freely through the basement membrane into the hemolymph (Harmansa et al, 2017; Ma et al, 2017), then as larvae mature, the hemolymph concentration of Dpp would be predicted to rise. It is therefore surprising that Dpp signaling in the PG decreases concurrently. One possibility is that the circulating levels of Dpp do indeed rise and the PG becomes less sensitive to Dpp, perhaps by down-regulating Dpp receptors or other signaling components. However, pulses of Dpp in late L3 using *rn*[ts]>*dpp* elicit strong reporter responses in the PG arguing against this possibility (data not shown). Although we are unable to reliably measure circulating physiological levels of native Dpp at different time points in L3, our data are most consistent with the possibility that circulating levels of Dpp do indeed decrease with time.

How could circulating Dpp levels decrease during L3? We present a speculative model (Fig 8) that might become directly testable as methods for detecting low levels of circulating Dpp improve. Although Dpp levels in discs increase during L3 (Wartlick et al, 2011), disc growth results in a disproportionate increase in the number of cells that do not produce Dpp, yet can bind Dpp, thus expanding the "morphogen sink". In addition, extracellular matrix proteins secreted by the fat body continue to be deposited on the basement membrane throughout L3 (Pastor-Pareja & Xu, 2011) and may lead to decreased permeability. For either or both of these reasons, as discs grow, more Dpp might be retained within the discs and less allowed to escape into the hemolymph and reach the PG (model in Fig 8). Indeed, reduced escape from discs could contribute to the observed increase in Dpp levels in discs as larvae progress through L3. As discs grow significantly during L3, organismal growth also leads to a dramatic size increase of the larval body, which has an open circulatory system as opposed to a vascular system. Therefore, it is possible that an increasing ratio of hemolymph to circulating Dpp is effectively causing a dilution of the ligand titer that correlates with growth. If indeed Dpp levels drop as a result of some or all of these reasons, then it provides a mechanism whereby the growth and maturation of the larva can influence the timing of pupariation.

Although a reduction in Dpp signaling in the PG seems necessary for the timely onset of metamorphosis, reducing Dpp signaling in the PG in well-fed larvae is insufficient to trigger entry into metamorphosis. This suggests that the timing of metamorphosis is primarily dictated by other signals such as the secretion of PTTH or by multiple signaling pathways working together. However, our finding that Dpp signaling seems necessary for the critical-weight checkpoint supports the notion that a decline in circulating Dpp levels might be a signal that indicates that sufficient tissue growth or maturation has occurred to generate a viable adult following metamorphosis. Several vertebrate BMP proteins have also been detected in the circulation (for example van Baardewijk et al [2013])

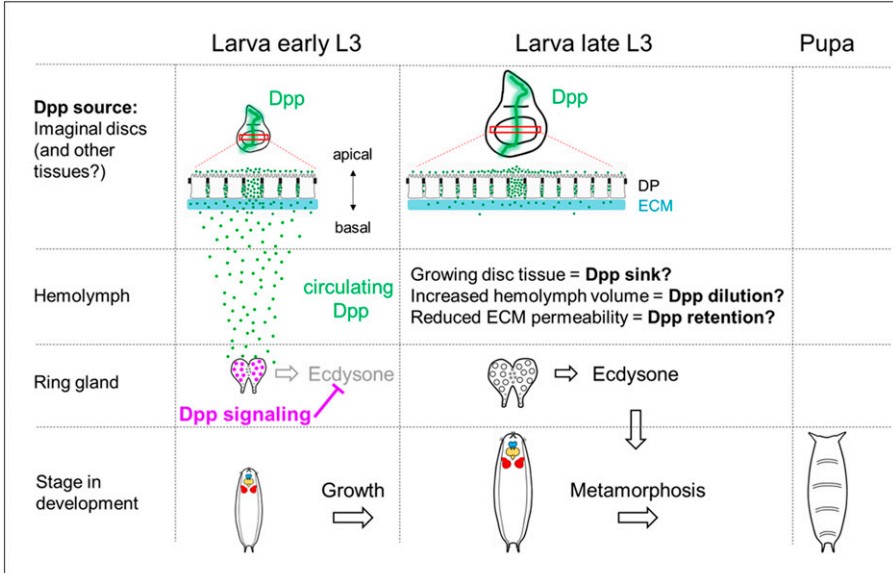

**Figure 8. Model of how Dpp from imaginal discs might regulate developmental timing.**
As the third instar larva matures, a large increase in imaginal disc size occurs while Dpp is continuously expressed in the imaginal disc. Surprisingly, the PG displays pMad activation early in L3 and its likely source is from the imaginal discs. pMad activity in the PG decreases and is absent late in L3. Because pMad inhibits ecdysone synthesis, a reduction in Dpp signaling in late L3 is necessary to produce sufficient ecdysone required for the timely onset of metamorphosis. This reduction of Dpp signaling may be due to growing disc tissue representing a sink that is outgrowing the source of Dpp, thereby retaining Dpp in discs at an increasing rate; or, an increase in hemolymph volume may outpace the secretion of Dpp, leading to a decreased concentration of circulating ligand that is no longer sufficient to activate pMad. It is also possible that continuous secretion of collagen from the fat body renders the ECM increasingly impermeable during L3, which would reduce the amount of Dpp that can be secreted from the disc.

and it will be of interest to know whether those proteins also have roles in regulating developmental timing.

## Materials and Methods

### *Drosophila* strains and husbandry

Animals were raised on standard medium as used by the Bloomington *Drosophila* Stock Center. Fly stocks used were $w^{1118}$ as wild-type control for all experiments, *Oregon-R, rn-Gal4,tub-Gal80$^{ts}$/TM6B-Gal80, rn-Gal4,tub-Gal80$^{ts}$,UAS-egr/TM6B-Gal80,* and *rn-Gal4,tub-Gal80$^{ts}$,UAS-rpr/TM6B-Gal80* (Smith-Bolton et al, 2009), *UAS-dpp/TM6B* (Haerry et al, 1998), *UAS-tkv$^{Q253D}$/TM6B* (Nellen et al, 1996), *dad-nRFP/CyO* (Wartlick et al, 2011), *UAS-GFP-Dpp* (Entchev et al, 2000), *UAS-dilp8:: 3xFLAG* (Garelli et al, 2012), *dpp-LG,* and *lexOEGFP::dpp/TM6B* (Yagi et al, 2010), *UAS-bantamA* and *bantam.sensor* (Brennecke et al, 2003), *P0206-Gal4, phm-Gal4,* and *ptth-Gal4* (from Lynn Riddiford), *UAS-tkv$^{D95K}$UAS-dad* and *UAS-brk* (from Christine Rushlow), *UAS-babo$^{CA}$* (from Michael O'Connor), *elav-Gal80* (Yang et al, 2009) (from Lily Yeh Jan and Yuh-Nung Jan), *dlg$^{40-2}$* (from David Bilder), and *dpp$^{FRT-PSB}$* (Bosch et al, 2017) (from Jean-Paul Vincent). Stocks that were obtained from the Bloomington *Drosophila* Stock Center: *dpp-lacZ* (#8404, #8411, #8412) *UAS-InR$^{ACT}$* (#8263), *dpp-Gal4/TM6B* (#1553), *UAS-dpp* (#1486), *UAS-tkv$^{CA}$* (#36537), *brk-GFP.FPTB* (#38629), *dilp8$^{MI00727}$* (#33079), *Aug21-Gal4* (#30137), *AKH-Gal4* (#25684), *UAS-dcr2* (#24650), *UAS-dpp$^{RNAi}$* (#25782), *r4-Gal4* (#33832), *UAS-preproANF-EMD* (#7001), *G-TRACE-3* (#28281), *UAS-GFP.nls* (#4775, #4776), *UAS-Mad$^{RNAi}$* (#31315, 43183), *UAS-tkv$^{RNAi}$* (#35653, #40937), *AbdB-Gal4* (#55848), *UAS-tdTom* (#36328), *UAS-tdGFP* (#35836), and *ap-GAD* (#54268). *UAS-Med$^{RNAi}$* (#19688) was from the Vienna *Drosophila* Resource Center (VDRC).

### Immunohistochemistry and microscopy

Larvae were dissected in PBS, fixed 20 min in 4% PFA, permeabilized with 0.1% Triton X-100, and blocked with 10% normal goat serum.

Primary antibodies used are: rabbit anti-Smad3 (phospho S423 + S425) (#52903, 1:500; Abcam), rabbit anti-Sad (1:250), rabbit anti-Phm (1:250), rabbit anti-Dib (1:250) and guinea pig anti-Spok (1:1000) (gifts from Michael O'Connor), rabbit anti-Ptth (1:100) (gift from Pierre Léopold), mouse anti-Armadillo N2 7A1 (Riggleman et al, 1990) (DSHB, 1:100), mouse anti-Dlg 4F3 (Parnas et al, 2001) (DHSB, 1:100), rabbit anti-GFP (#TP401, 1:500; Torrey Pines Biolabs), mouse anti-GFP (AB290, 1:500; Abcam), rabbit anti-cleaved DCP-1 (Asp216, 1:250; Cell Signaling Technology), and rabbit anti-FOXO (1:1000) (gift from Michael Thomas Marr) (Puig et al, 2003). Secondary antibodies used are: goat anti-mouse 555 (#A32727; Invitrogen), goat anti-mouse 647 (#A32728; Invitrogen), goat anti-rabbit 555 (#A32732; Invitrogen), goat anti-rabbit 647 (#A32733; Invitrogen), goat anti-guinea pig 555 (#A-21435; Invitrogen) (all, 1:500), as well as phalloidin-TRITC (#P1951, 1:500; Sigma-Aldrich) and DAPI (#D1306, 1:500; Invitrogen). Samples were mounted in SlowFade Gold (#S36937; Invitrogen) and imaged on a Zeiss 700 LSM confocal microscope.

### Developmental timing assay and *rn$^{ts}$>* temperature shift experiments

Fertilized eggs were collected on grape juice plates for 4 h. L1 stage larvae were transferred onto standard Bloomington food supplemented with yeast paste at a density of 50 animals per vial. For constitutive expression without the presence of a temperature-sensitive *Gal80,* animals were raised consistently at 25°C and pupal counts were taken every 8 h. Three independent experiments were conducted for each condition. *rn$^{ts}$>* animals were raised at 18°C until day 7 (early third instar), then transferred to 30°C for a 24 h temperature shift and subsequently returned to 18°C. Pupal counts were taken every 12 h. Three independent experiments were conducted for each condition. In the graphs, error bars show standard deviations between the experiments and n stands for the total number of pupae that were counted.

**Table 1.** Statistical analysis of pupal weight of each genotype versus every other genotype in Fig 7N.

| Tukey's multiple comparisons test | P Value |
|---|---|
| $P0206>$ versus $P0206>tkv^{QD}$ | 0.0001 to 0.001 |
| $P0206>$ versus $P0206>InR^{ACT}$ | 0.0001 to 0.001 |
| $P0206>$ versus $P0206>InR^{ACT}+tkv^{QD}$ | ≥0.05 |
| $P0206>$ versus $P0206>babo^{CA}$ | <0.0001 |
| $P0206>$ versus $P0206>babo^{CA}+tkv^{QD}$ 40% | <0.0001 |
| $P0206>$ versus $P0206>babo^{CA}+tkv^{QD}$ 60% | ≥0.05 |
| $P0206>$ versus $P0206>banA$ | 0.0001 to 0.001 |
| $P0206>$ versus $P0206>banA+tkv^{QD}$ | <0.0001 |
| $P0206>tkv^{QD}$ versus $P0206>InR^{ACT}$ | <0.0001 |
| $P0206>tkv^{QD}$ versus $P0206>InR^{ACT}+tkv^{QD}$ | ≥0.05 |
| $P0206>tkv^{QD}$ versus $P0206>babo^{CA}$ | <0.0001 |
| $P0206>tkv^{QD}$ versus $P0206>babo^{CA}+tkv^{QD}$ 40% | <0.0001 |
| $P0206>tkv^{QD}$ versus $P0206>babo^{CA}+tkv^{QD}$ 60% | ≥0.05 |
| $P0206>tkv^{QD}$ versus $P0206>banA$ | ≥0.05 |
| $P0206>tkv^{QD}$ versus $P0206>banA+tkv^{QD}$ | <0.0001 |
| $P0206>InR^{ACT}$ versus $P0206>InR^{ACT}+tkv^{QD}$ | <0.0001 |
| $P0206>InR^{ACT}$ versus $P0206>babo^{CA}$ | <0.0001 |
| $P0206>InR^{ACT}$ versus $P0206>babo^{CA}+tkv^{QD}$ 40% | <0.0001 |
| $P0206>InR^{ACT}$ versus $P0206>babo^{CA}+tkv^{QD}$ 60% | <0.0001 |
| $P0206>InR^{ACT}$ versus $P0206>banA$ | <0.0001 |
| $P0206>InR^{ACT}$ versus $P0206>banA+tkv^{QD}$ | <0.0001 |
| $P0206>InR^{ACT}+tkv^{QD}$ versus $P0206>babo^{CA}$ | <0.0001 |
| $P0206>InR^{ACT}+tkv^{QD}$ versus $P0206>babo^{CA}+tkv^{QD}$ 40% | <0.0001 |
| $P0206>InR^{ACT}+tkv^{QD}$ versus $P0206>babo^{CA}+tkv^{QD}$ 60% | ≥0.05 |
| $P0206>InR^{ACT}+tkv^{QD}$ versus $P0206>banA$ | ≥0.05 |
| $P0206>InR^{ACT}+tkv^{QD}$ versus $P0206>banA+tkv^{QD}$ | <0.0001 |
| $P0206>babo^{CA}$ versus $P0206>babo^{CA}+tkv^{QD}$ 40% | ≥0.05 |
| $P0206>babo^{CA}$ versus $P0206>babo^{CA}+tkv^{QD}$ 60% | <0.0001 |
| $P0206>babo^{CA}$ versus $P0206>banA$ | <0.0001 |
| $P0206>babo^{CA}$ versus $P0206>banA+tkv^{QD}$ | <0.0001 |
| $P0206>babo^{CA}+tkv^{QD}$ 40% versus $P0206>babo^{CA}+tkv^{QD}$ 60% | <0.0001 |
| $P0206>babo^{CA}+tkv^{QD}$ 40% versus $P0206>banA$ | <0.0001 |
| $P0206>babo^{CA}+tkv^{QD}$ 40% versus $P0206>banA+tkv^{QD}$ | <0.0001 |
| $P0206>babo^{CA}+tkv^{QD}$ 60% versus $P0206>banA$ | ≥0.05 |
| $P0206>babo^{CA}+tkv^{QD}$ 60% versus $P0206>banA+tkv^{QD}$ | <0.0001 |
| $P0206>banA$ versus $P0206>banA+tkv^{QD}$ | <0.0001 |

## Quantification of wing discs and pupal weight

Pupae from developmental timing assays collected at the pharate adult stage were cleaned with 70% ethanol, dried, and weighed in groups of 30 in three or four independent experiments. In the graphs, error bars show standard deviations between experimental groups and n stands for the total number of pupae that were weighed. GraphPad Prism 6 was used to determine statistical significance between groups by one-way ANOVA using Tukey's or Dunnett's test. Pupae were placed on double-sided adhesive tape for imaging using a Leica transmitted light microscope (TL RCI). Adobe Photoshop was used to quantify the area of imaginal disc confocal images dissected from larvae from developmental timing assays.

## Western blotting

Wing imaginal discs were dissected in chilled PBS supplemented with protease inhibitor (#11697498001; Roche) and directly transferred into Laemmli sample buffer, then boiled for 10 min. Hemolymph was extracted by bleeding larvae into chilled PBS supplemented with protease inhibitor on a cold aluminum block using a fine tungsten needle to puncture the cuticle, then transferred into Laemmli sample buffer and boiled for 10 min. Samples were run on 10% Mini-Protean TGX gels (Bio-Rad) and transferred to nitrocellulose membrane (Bio-Rad). The primary antibody used was rabbit anti-GFP (#TP401, 1:1,000; Torrey Pines Biolabs). Protein bands were detected with secondary antibody HRP anti-rabbit (#sc-2030, 1:2,500; Santa Cruz Biotechnology), and Western Lightning Plus-ECL (#NEL103001EA; PerkinElmer).

## Ex vivo organ culture

20 brain-ring gland complexes were dissected from wandering third instar Oregon R larvae in Schneider's medium (#21720024; Gibco) by pulling mouth hooks from which salivary glands, lymph glands, and fat body were removed. Complexes were subsequently co-cultured for 3 h with either wing imaginal discs or hemolymph from $rn^{ts}>dpp$ larvae larvae in Schneider's medium supplemented with 10% FBS (#26140087; Invitrogen) and penicillin-streptomycin at 1:100 of a 5,000 U/ml stock (#15070063; Gibco).

## Irradiation

Density controlled third instar larvae were placed on shallow food plates and irradiated with 45 Gy in an X-ray cabinet (Faxitron), followed by dissection after 12 h.

## Heat-shock clone induction

Flies with *UAS* transgenes were crossed to *ywhsFlp;;Act≫Gal4, UAS-GFP* and raised at 25°C. Larvae were staged and density controlled as described for developmental timing assay, then heat shocked in a 37°C water bath for 5 min at 24 or 48 h AEL before returning to 25°C until dissection.

## Ecdysone feeding

L1 stage larvae were transferred onto standard Bloomington food supplemented with yeast paste at a density of 50 animals per vial. 1 mg 20-hydroxyecdysone (Sigma-Aldrich) in ethanol (100 *µ*l of a 10 mg/ml solution) was added to the surface of each food vial.

## Developmental staging and starvation assay

Before egg collection, flies were transferred to a constant light environment for at least 2 d and all subsequent treatments were carried out under constant light to avoid possible influences from diuranal cycles. Eggs were collected on apple juice plates with yeast paste for 4 h and early L1 larvae were transferred to standard laboratory fly food with yeast paste after hatching. After larvae developed to the late L2 stage, newly molted L3 larvae were picked out every 2 h and transferred to new fly food without yeast paste. For the starvation assay, L3 larvae were cultured for the appropriate time and then transferred to 1% agar for starvation. Larvae were then monitored every 2 h until they pupariated or died.

# Supplementary Information

# Acknowledgements

We thank many colleagues in the fly community for constructive suggestions; María Domínguez, Marcos González-Gaitán, Pierre Léopold, Michael O'Connor, Lynn Riddiford, Chris Rushlow, Michael Thomas Marr, Hilary Ashe, Jean-Paul Vincent, Markus Affolter, and Konrad Basler for fly stocks and antibodies; the Bloomington, VDRC and TRiP stock centers; Octavio Bejarano, Jane Thomas, and Lupita Hernandez for technical assistance; and Jo Downes Bairzin, David Bilder, Robin Harris, Nipam Patel, Taryn Sumabat, and Melanie Worley for comments on the manuscript. IK Hariharan was funded by National Institutes of Health (NIH) grant R35GM122490 and an American Cancer Society Research Professor Award (RP-16-238-06-COUN). MB O'Connor was funded by NIH grant R35GM118029.

## Author Contributions

IK Hariharan: conceptualization, supervision, funding acquisition, and writing—original draft, project administration, review and editing.
L Setiawan: conceptualization, formal analysis, investigation, visualization, methodology, and writing—original draft, review, and editing.
AL Woods: data curation and investigation.
MB O'Connor: conceptualization, supervision, and writing—review and editing.
X Pan: conceptualization, investigation, visualization, and writing—review and editing.
All data with the exception of Fig 5F–L were generated by L Setiawan with assistance from AL Woods. Data in Fig 5F–L were generated by X Pan.

## Conflict of Interest Statement

The authors declare no conflicts of interest.

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
