## [Reviewer comments · Life Science Alliance]

Life Science Alliance

The BMP2/4 ortholog Dpp can function as an inter-organ signal that regulates developmental timing

Iswar Hariharan, Linda Setiawan, Alexis Woods, Michael O'Connor, and Xueyang Pan
DOI: 10.26508/lsa.201800216

Corresponding author(s): Iswar Hariharan, Univ. of California, Berkeley

Review Timeline:	Submission Date:	2018-10-15
	Editorial Decision:	2018-10-16
	Revision Received:	2018-11-06
	Accepted:	2018-11-07

Scientific Editor: Andrea Leibfried

Transaction Report:

Please note that the manuscript was previously reviewed at another journal and the reports were taken into account in inviting a revision for publication at *Life Science Alliance* prior to submission to *Life Science Alliance*.

1st Editorial Decision

16 October 2018

Thank you for submitting your revised manuscript entitled "The BMP2/4 ortholog Dpp can function as an inter-organ signal that regulates developmental timing" to Life Science Alliance. Your manuscript was previously re-reviewed by an arbitrator at another journal, and the editors transferred those comments to us with your permission.

The arbitrating advisor pointed out that your work is of high quality. However, the advisor also noted that while you satisfactorily addressed most concerns of the reviewers, the concern remains that your data and conclusions rely on Dpp overexpression experiments. The

biological significance of your findings therefore remains somewhat unclear. This is not a concern for publication in Life Science Alliance, and we would be happy to accept your paper pending final revisions necessary to meet our formatting guidelines.

REFEREE REPORTS OBTAINED DURING PEER REVIEW ELSEWHERE

Referee #1 Review

Report for Author:

The present ms identifies the ability of the signalling molecule Dpp, well known to regulate in a local manner the pattern and growth of the developing larval primordia, to act in a systemic manner, signal to the Ecdysone-producing organ (prothoracic gland), regulate Ecdysone production and delay developmental timing. The ms is well written, figures are self-explanatory and absolutely all experiments to demonstrate the above message have been well performed. I do not think I have anything else to add at this point as previous reviewers have done an excellent work in raising the most important points and authors have addressed them satisfactorily.

However, as pointed by those reviewers, the paper is based only on Dpp overexpression experiments and, unfortunately, Dpp depletion in larval primordia does not appear to have any consequence on developmental timing (larval/pupal transition). To be fair with my previous colleagues, I conclude that authors have not satisfactorily addressed this important issue, as the biological significance of their findings remains to be elucidated.

2nd Editorial Decision

7 November 2018

Thank you for submitting the final version of your manuscript "The BMP2/4 ortholog Dpp can function as an inter-organ signal that regulates developmental timing" to Life Science Alliance. I reviewed the new data you and your colleagues added and I appreciate that you now additionally demonstrate a role for Dpp signaling in controlling the critical weight checkpoint and pupariation under limited food availability. It is a pleasure to let you know that your manuscript is now accepted for publication in Life Science Alliance.

Congratulations on this interesting work.